# Association of Neutrophil, Platelet, and Lymphocyte Ratios with the Prognosis in Unresectable and Metastatic Pancreatic Cancer

**DOI:** 10.3390/jcm9103283

**Published:** 2020-10-13

**Authors:** Jessica Allen, Colin Cernik, Suhaib Bajwa, Raed Al-Rajabi, Anwaar Saeed, Joaquina Baranda, Stephen Williamson, Weijing Sun, Anup Kasi

**Affiliations:** 1Department of Internal Medicine, School of Medicine, University of Kansas, 3901 Rainbow Blvd, Kansas City, KS 66160, USA; jallen19@kumc.edu (J.A.); sbajwa@kumc.edu (S.B.); 2Department of Biostatistics and Data Science, University of Kansas, 1450 Jayhawk Blvd, Lawrence, KS 66045, USA; ccernik@kumc.edu; 3University of Kansas Cancer Center, 2650 Shawnee Mission Pkwy, Westwood, KS 66205, USA; ral-rajabi@kumc.edu (R.A.-R.); asaeed@kumc.edu (A.S.); jbaranda@kumc.edu (J.B.); swilliam@kumc.edu (S.W.); wsing2@kumc.edu (W.S.)

**Keywords:** pancreatic cancer, pancreatic ductal adenocarcinoma, pancreatic cancer prognosis

## Abstract

We examined the relationship between the daily rate of change of cancer antigen 19-9 (CA19-9) over the first 90 days of treatment (DRC90) and the pretreatment levels of neutrophils, lymphocytes, and platelets with the overall survival (OS) and progression-free survival (PFS) in patients with stage IV pancreatic ductal adenocarcinoma (PDA) who received chemotherapy. We retrospectively evaluated 102 locally advanced and metastatic PDA patients treated at the University of Kansas Cancer Center (KUCC) between January 2011 and September 2019. We compared the ratio of the pretreatment absolute neutrophil count to the pretreatment absolute lymphocyte count (NLR) and the ratio between the pretreatment platelet count to the pretreatment absolute lymphocyte count (PLR) with the OS and PFS. We compared the DRC90 to the OS and PFS. The ratios were analyzed using the log-rank trend test using the mean of the NLR, PLR, and DRC90 as the threshold for two groups within each variable. Patients with ≥mean NLR (4.6 K/µL) had a significantly lower OS (*p* = 0.0444) and PFS (*p* = 0.0483) compared with patients below the mean. Patients with PLR ≥ mean (3.9 K/µL) did not have a significantly different OS (*p* = 0.507) or PFS (*p* = 0.643) compared with patients below the mean. Patients with DRC90 ≥ mean (−1%) did not have a significantly different OS (*p* = 0.342) or PFS (*p* = 0.313) compared with patients below the mean. Patients with NLR ≥ mean (4.6 K/µL) had a significantly lower OS and PFS compared with patients with NLR below the mean. This implies the possibility of NLR as a prognostic marker in PDA that could guide treatment approaches but still requires validation in a larger cohort.

## 1. Introduction

Pancreatic adenocarcinoma is the third leading cause of death from cancer in the United States with a 5-year survival rate of 9% [1]. There currently no known sufficiently sensitive methods of screening asymptomatic adults for pancreatic adenocarcinoma, and thus, the United States Preventive Services Task Force (USPSTF) currently recommends against screening until symptoms develop [2]. Patients are often diagnosed at later stages, because pancreatic cancer is commonly asymptomatic in the early stages [1]. The effects of this are grim: in patients with stage IV pancreatic adenocarcinoma, the 5-year survival rate drops to 3% [1]. The survival of pancreatic adenocarcinoma has not improved substantially in forty years [3]. Additionally, few risk factors or markers of prognosis have been identified [3].

Individual studies have examined the effect of lymphocytes, neutrophils, and platelets on the overall survival (OS) and progression-free survival (PFS) in patients with pancreatic adenocarcinoma. One study found that a ratio between the pretreatment platelet count to the pretreatment absolute lymphocyte count (PLR) of >240 K/µL, a ratio of the pretreatment absolute neutrophil count to the pretreatment absolute lymphocyte count (NLR) > 5 K/µL, and a daily rate of change of CA19-9 over the first 90 days of treatment (DRC90) > 0.4% were significantly associated with a decreased OS [4]. Another study confirmed that an elevated NLR at or above 3.54 K/µL was significantly associated with a decreased OS [5]. An additional study contradictorily found that NLR > 5 K/µL and CA19-9 at the time of diagnosis ≥ 437 µ/mL were significantly associated with an increased OS and PFS [6]. Further analysis of the effects of the amounts and ratios of immune cells and the rate of tumor marker change in patients at the time of diagnosis on the overall survival is required to ascertain their utility in predicting prognosis and guiding treatment. Thus, in this retrospective study, we analyzed the effect of the NLR, PLR, and DRC90 on the OS and PFS in locally advanced and metastatic pancreatic cancer patients treated at the University of Kansas Cancer Center to add to the existing knowledge regarding the association immune cell ratios have with the prognosis in pancreatic adenocarcinoma.

## 2. Experimental Section

The study was a retrospective chart review of the characteristics of patients with locally advanced or metastatic pancreatic adenocarcinoma diagnosed between January 2011 and September 2019 from the University of Kansas Cancer Center medical records.

The primary outcomes of interest were the OS and PFS of patients with locally advanced or metastatic pancreatic ductal carcinoma. The NLR and PLR were compared with the OS and PFS. The complete blood cell count closest to the date of the initiation of treatment was used for the neutrophil, lymphocyte, and platelet levels, and patients without a pre-treatment complete blood count (CBC) available were excluded. The DRC90 was calculated and compared with the OS and PFS. The CA19-9 at the time of diagnosis was used as the baseline for measuring the rate of change. The DRC90 was found by calculating the daily percent change of the baseline CA19.9 and the CA19.9 at three months after the treatment initiation. Demographic data, such as the age, gender, race, smoking status, Eastern Cooperative Oncology Group (ECOG) performance status, tumor location, and treatment received, were also collected. All patient data was collected retrospectively via electronic health records and stored on a secure Redcaps database. 

The data were deidentified before analysis. Associations between the NLR, PLR, and DRC90 and the OS and PFS were obtained via Kaplan–Meier survival curves and log-rank trend tests using the means of NLR (4.6), PLR (196), and DRC90 (−1%) as the threshold for the two groups within each variable. 

## 3. Results

### 3.1. Characteristics of Patients

A total of 102 patients diagnosed with locally advanced or metastatic pancreatic cancer with pretreatment complete blood cell counts and pretreatment CA19-9 available, diagnosed between January 2011 and September 2019 from the University of Kansas Cancer Center were included in the study. Patients were split into two groups based on whether they fell at or above the mean NLR, mean PLR, and mean DRC90 or below. The patient characteristics within each group are listed in Table 1.

### 3.2. Efficacy

The median progression-free survival for patients with an NLR greater to or equal than the population’s mean (4.6 K/µL) was 259 days (95% CI 177–308 days), and the median overall survival was 387 days (95% CI 221–455 days). For patients below the sample’s mean NLR (<4.6 K/µL), the median PFS was 339 days (95% CI 207–592 days), and the median OS was 491 days (95% CI 391–527 days). The log-rank test showed that the PFS and OS for patients with an NLR greater than or equal to the mean of 4.6 K/µL and patients with an NLR less than 4.6 K/µL were significantly different (*p* = 0.0444, *p* = 0.0483, respectively), which is shown in Figure 1a,b. An elevated NLR in comparison to the sample population mean was associated with a lower OS and PFS in locally advanced or metastatic pancreatic adenocarcinoma.

The median PFS for patients above the sample’s PLR mean of 196 was 267 days (95% CI 177–361 days) and the median OS was 425 days (95% CI 302–494 days). The median PFS for patients below the samples PLR mean was 314 days (95% CI 187–484 days), and the median OS was 477 days (95% CI 381–527 days). The log-rank test did not show significant differences in OS and PFS in patients with a PLR greater than or equal to the sample mean and PLR below the sample mean, as shown in Figure 1c,d.

The median PFS for patients with a DRC90 greater than or equal to the sample mean (−1%) was 176 days (95% CI 101–368 days), and the median OS was 414 days (95% CI 279–547 days). The median PFS for patients with a DRC90 less than the sample mean was 308 days (95% CI 258–455 days), and the median OS was 451 days (95% CI 387–521 days). There was no significant difference in the OS and PFS between the DRC90 groups above the sample mean of −1% and below it, as shown in Figure 1e,f.

## 4. Discussion

Our findings of a significant association between an elevated NLR and lower OS and PFS agree with the findings of Das et al. and Dede et al., and are contradictory to the findings of Desai et al [4,5,6]. Thus, three separate studies have now found a significant association between an elevated NLR and lower OS and PFS in patients with locally advanced or metastatic disease, with a combined 283 patients studied. The cutoff NLR value of 4.6 K/µL used to separate groups in this study was lower than the NLR cutoff value of 5 K/µL used by Das et al. and higher than the NLR cutoff value of 3.54 K/µL used by Dede et al. [4,5]. Further investigation into the exact threshold of NLR elevation at which the OS and PFS is negatively affected is warranted in order to guide the treatment of patients with an elevated NLR, as our study further implicates elevated NLR as a negative prognostic factor.

A numerical difference in the PFS and OS was found between the PLR groups, with groups below the mean of 196 showing a PFS of 314 days and an OS of 477 days versus a PFS of 267 and an OS of 425 days in the groups above the sample mean PLR. Although statistical significance was not met, this numerical difference showing a lower PFS and OS in patients with PLR below the sample mean agrees with the statistically significant findings of Das et al., who found an association between an elevated PLR and lower OS and PFS [4].

A numerical difference in the PFS and OS was found between the DRC90 groups, with groups below the mean DRC90 of −1% showing a PFS of 308 days and OS of 451 days versus PFS of 176 and OS of 414 days in the group greater than the mean. Although a statistical difference was not met, this numerical trend agrees with the findings of Das et al. [4].

Our finding of elevated NLR as a poor marker of prognosis agrees with studies done on other forms of cancer [7]. Neutrophils are key components of the innate immune response but have also been implicated in the biogenesis of malignancy via their ability to blunt antitumor T cell responses [8]. Specific subsets of mature neutrophils have shown to play an important role in the escape of tumor cells from antitumor immunity [9]. The current treatment guidelines for metastatic pancreatic adenocarcinoma are largely based on the ECOG status and comorbidity profile [10]. Evidence of the importance of elevated NLR on the outcomes in metastatic pancreatic adenocarcinoma that our study and others provide, as well as larger studies including multiple forms of malignancy, indicate the potential for baseline NLR at the time of diagnosis to factor into treatment guidelines.

The limitations of this study include the retrospective chart review design, the small sample size of analysis (*n* = 102), variability in the treatment regimens received by patients, possible homogeneity due to the sample selection from only one treatment center, and group analysis of both metastatic and locally advanced patients. Our analysis grouped metastatic and locally advanced patients due to sample size limitations, and future studies should analyze these cohorts separately to assess the impact of distant metastasis versus locally advanced disease on outcomes in patients with an elevated NLR.

A large prospective randomized study is needed to confirm our study results, to ascertain the utility of elevated NLR as a prognostic marker in pancreatic cancer, and to further determine what threshold of elevation is universally prognostic. Given the results of our study, which used a threshold NLR of 4.6 K/µL to separate groups, along with the results of Des et al. with their threshold of 5 K/µL and the threshold of 3.54 K/µL used by Dede et al., future studies might start with a threshold below that of Dede et al. to further ascertain at what point an elevated NLR impacts outcomes.

## Figures and Tables

**Figure 1 jcm-09-03283-f001:**
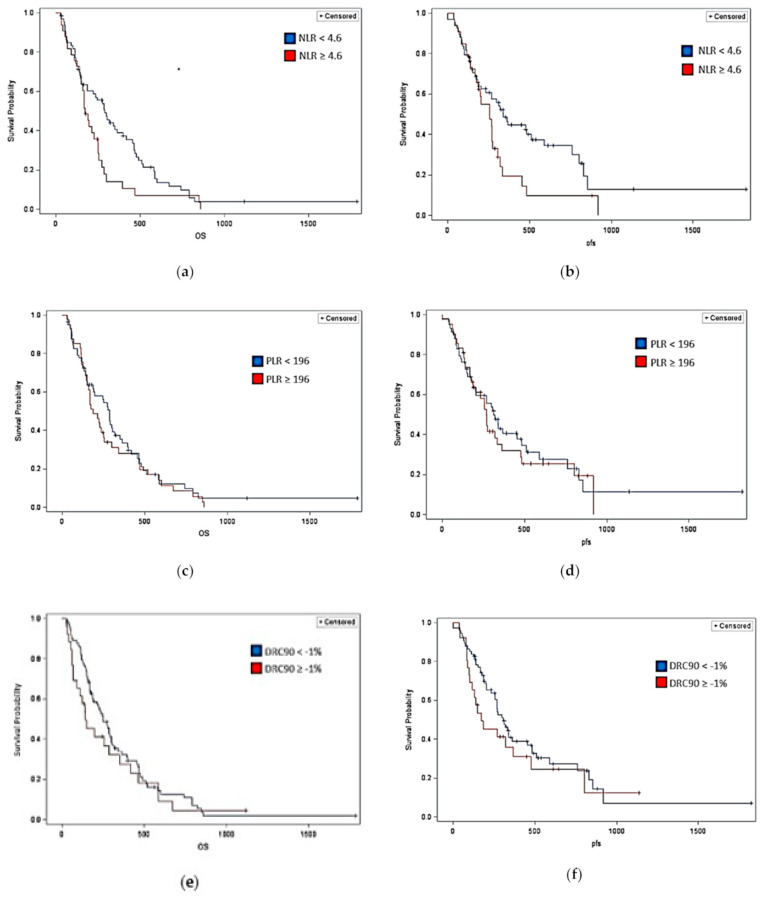
(**a**) Kaplan–Meier survival curve comparison of the OS of NLR groups; (**b**) Kaplan–Meier survival curve comparison of the PFS of NLR groups; (**c)** Kaplan–Meier survival curve comparison of the OS of PLR groups; (**d**) Kaplan–Meier survival curve comparison of the PFS of PLR groups; (**e**) Kaplan–Meier survival curve comparison of the OS of DRC90 groups; (**f**) Kaplan–Meier survival curve comparison of the PFS of DRC90 groups.

**Table 1 jcm-09-03283-t001:** Characteristics of the patients in each group of analysis. The ratio between the pretreatment platelet count to the pretreatment absolute lymphocyte count (PLR), the ratio of the pretreatment absolute neutrophil count to the pretreatment absolute lymphocyte count (NLR), and the daily rate of change of CA19-9 over the first 90 days of treatment (DRC90).

Characteristics	NLR < 4.6 K/µL	NLR ≥ 4.6 K/µL	PLR < 196 K/µL	PLR ≥ 196 K/µL	DRC90 < −1%	DRC90 ≥ −1%
Number	66	35	59	43	66	36
Age (median)	65.5	62	63	64	64	61.5
Gender (%)						
Male	62.0%	60.0%	61.0%	62.8%	60.6%	63.9%
Female	38.0%	40.0%	39.0%	37.20%	39.4%	36.1%
Race						
White	57 (86.4%)	32 (91.4%)	53 (89.8%)	36 (83.7%)	60 (90.9%)	29 (80.6%)
Black or African American	2 (3.0%)	2 (5.7%)	2 (3.4%)	2 (4.7%)	0 (0.0%)	4 (11.1%)
Other	5 (7.6%)	1 (2.9%)	4 (6.8%)	5 (11.6%)	6 (9.1%)	3 (8.3%)
Smoking Status						
Yes	27 (40.9%)	17 (48.6%)	30 (50.8%)	14 (32.6%)	40 (60.6%)	19 (52.8%)
No	37 (56.1%)	17 (48.6%)	27 (45.8%)	28 (65.1%)	25 (37.9)	15 (41.7%)
ECOG Status						
0–1	60 (90.1%)	33 (94.3%)	54 (91.5%)	39 (90.7%)	60 (90.9%)	33 (91.7%)
2 or higher	6 (9.1%)	2 (5.7%)	5 (8.5%)	4 (9.3%)	5 (7.6%)	2 (5.6%)
Tumor Location						
Head	49 (74.2%)	15 (42.3%)	40 (67.8%)	24 (55.8%)	42 (63.6%)	22 (61.1%)
Tail	9 (13.6%)	11 (31.4%)	10 (16.9%)	6 (14.0%)	16 (24.3%)	5 (13.9%)
Body	7 (10.6%)	9 (25.7%)	8 (13.6%)	13 (30.2%)	7 (10.6%)	9 (25.0%)
Neck	1 (1.5%)	0 (0.0%)	1 (1.7%)	0 (0.0%)	1 (1.5%)	0 (0.0%)
Metastatic	46 (69.7%)	33 (94.3%)	46 (78.0%)	34 (79.0%)	54 (81.8%)	33 (91.7%)
Locally Advanced	20 (30.3%)	2 (5.7%)	13 (22.0%)	9 (21.0%)	12 (18.2%)	3 (8.3%)
CA19-9 at the time of diagnosis						
Normal (<38 U/mL)	11 (16.7%)	3 (8.6%)	8 (13.6%)	5 (11.6%)	5 (7.6%)	8 (22.2%)
Abnormal (>38 U/mL)	54 (81.8%)	32 (91.4%)	51 (86.4%)	38 (88.4%)	61 (92.4%)	26 (72.2%)
Treatment Received						
FOLFIRINOX	40 (60.6%)	20 (57.1%)	37 (62.7%)	24 (55.8%)	40 (60.6%)	21 (58.3%)
Gemcitabine/albumin-bound Paclitaxel (Abraxane)	16 (24.2%)	11 (31.4%)	13 (22.0%)	14 (32.6%)	18 (27.3%)	10 (27.8%)
Gemcitabine	4 (6.1%)	0 (0.0%)	4 (6.8%)	0 (0.0%)	2 (3.0%)	2 (5.6%)
Other	6 (9.1%)	4 (11.4%)	5 (8.5%)	5 (11.6%)	6 (9.1%)	3 (8.3%)

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
