# Peer review of "Association of Neutrophil, Platelet, and Lymphocyte Ratios with the Prognosis in Unresectable and Metastatic Pancreatic Cancer"

_jcm, 2020, doi:10.3390/jcm9103283_

Round 1

Reviewer 1 Report

Comments:

No description regarding the metastatic status of the patients, whether local or distant, provided in the result section. How the study performed in context of CA19.9 is not explained. Study lacks rigorous statistical analysis. 

Author Response

Response to Reviewer 1 Comments

We would like to sincerely thank you for reviewing our manuscript, and for providing important comments. We have taken all of the reviewer’s comments into consideration and have revised the manuscript accordingly. Below, please find our point-by-point responses to the reviewer’s comments.

Point 1: No description regarding the metastatic status of the patients, whether local or distant, provided in the result section.

Response 1: We appreciate this recommendation. We agree that more detailed subgroup analyses are needed. We have now included metastatic status in Table 1. We analysed both groups together in order to increase the power of the analysis due to low sample size, which was a major limitation in our analysis. example, there are only 2 locally advanced patients with NLR >4.6, as seen in Table 1, therefore we chose not to perform a subgroup analysis on advanced unresectable patients in order to preserve study power. We have included a statement in our discussion section on why we grouped the two cohorts in our analysis and why this is a limitation on the study (lines 182-185). we will take this comment into consideration when we plan future prospective trials to further evaluate the data from this present hypothesis-generating study.

Point 2: How the study performed in context of CA19.9 is not explained. Study lacks rigorous statistical analysis. 

Response 2: Statistical analysis was performed on the daily rate of change of CA19.9 in the first 90 days of treatment (DRC90). We measured this because another study found significance between higher DRC90 and decreased OS and PFS, and we wanted to test this in our cohort. Our analysis and the results of this analysis for DRC90 can be seen at lines 98-102. The median PFS for patients with DRC90 greater than or equal to the sample mean (-1%) was 176 days (95% CI 101-368 days) and the median OS was 414 days (95% CI 279-547 days). The median PFS for patients with DRC90 less than the sample mean was 308 days (95% CI 258-455 days) and the median OS was 451 days (95% CI 387-521 days). There was no significant difference in OS and PFS between the DRC90 groups, as shown in figure 1e and 1f. The Kaplan-Meier survival curve for DRC90 is also shown in figure 1. Additionally, in the discussion we highlight the numerical difference that was seen between the OS and PFS of the DRC90 groups (lines 167-170).  We have clarified how DRC90 was calculated (lines 66-67).

Reviewer 2 Report

Proofreading of the entire manuscript is an issue. A few repeated corrections to address: (1) the abbreviations for OS, PFS, NLR, PLR, DRC90 are defined more than once (2) write out or define pts as abbreviation for patients, and do not capitalize (3) put units on lab values (4) table formatting for various titles is inconsistent (5) CI units are inconsistently included. This is not exhaustive, there are scattered, single-errors as well to correct.

The first paragraph of the introduction is not part of the manuscript but is the instructions. Please delete.

Based on the introduction but also the title, it is suggested that this review considers patients with metastatic disease. However, in the experimental section and thereafter, patients with non-metastatic unresectable disease are also included. It is reasonable to consider unresectable patients in this incurable cohort. However, please revise to be consistent and introduce the locally advanced cohort as well. I also suggest to quantify how many of each category were studied. 

Is there an ability to consider the different outcomes based upon either locally advanced versus metastatic disease? I realize this may not be possible due to small numbers, but please consider commenting on this point.

The first paragraph of the discussion repeats a lot of the results around NLR and survival. This could be reserved to the results and thus would make for a more impactful discussion.

Given the findings here, I think it could be impactful to make a statement of how to use NLR in stratifying patients, noting of course that larger data are necessary. If, for example, the trial that the authors suggest were to go forward, how would they suggest stratifying around NLR? At what numbers?

I disagree with the statement of evaluating NLR as it relates to immunotherapy in pancreas cancer in the discussion, because immunotherapy has no role for treatment of pancreas cancer save for rare cases of MSI or on clinical trial. 

Author Response

Response to Reviewer 2 Comments

We would like to sincerely thank you for reviewing our manuscript, and for providing important comments. We have taken all of the reviewer’s comments into consideration and have revised the manuscript accordingly. Below, please find our point-by-point responses to the reviewer’s comments.

Point 1: Proofreading of the entire manuscript is an issue. A few repeated corrections to address: (1) the abbreviations for OS, PFS, NLR, PLR, DRC90 are defined more than once (2) write out or define pts as abbreviation for patients, and do not capitalize (3) put units on lab values (4) table formatting for various titles is inconsistent (5) CI units are inconsistently included. This is not exhaustive, there are scattered, single-errors as well to correct.

 Response 1:

  • The excess abbreviation definitions have been removed, as tracked on the word document of the manuscript. They are now defined only in the abstract, lines 12, 13, 16, and 17.
  • We have removed the “pts” abbreviation, as this was a relic from a word count requirement for a conference this study was presented at.
  • Treatment received was underlined in the table so that now all subcategories of patient demographics are underlined.
  • CI units are now included in all listings of CI, which can be seen throughout the results section and was tracked via the word document.

Point 2: The first paragraph of the introduction is not part of the manuscript but is the instructions. Please delete.

Response 2: This paragraph has now been deleted, as tracked in the word document.

Point 3: Based on the introduction but also the title, it is suggested that this review considers patients with metastatic disease. However, in the experimental section and thereafter, patients with non-metastatic unresectable disease are also included. It is reasonable to consider unresectable patients in this incurable cohort. However, please revise to be consistent and introduce the locally advanced cohort as well. I also suggest to quantify how many of each category were studied. 

Response 3: We have now included “unresectable” in the title, and have added “locally advanced” to the introduction, as seen on line 51. Additionally, Table 1 now includes quantification of metastatic and locally advanced subjects within each group.

Point 4: Is there an ability to consider the different outcomes based upon either locally advanced versus metastatic disease? I realize this may not be possible due to small numbers, but please consider commenting on this point.

Response 4: We appreciate this recommendation. We agree that more detailed subgroup analyses are needed. A major limitation in our analysis was the small sample size that precluded further stratification by subgroups. We acknowledge this limitation in the discussion section of the manuscript. For example, there are only 2 patients with locally advanced unresectable disease with NLR > 4.6, as seen in Table 1.Therefore, we chose not to perform a subgroup analysis in the locally advanced unresectable patient population in order to preserve power to observe any differences. In our limitations section, we included a statement that the interpretation of our findings with respect to of locally advanced unresectable patients is limited by the small sample size (lines 182-185). Certainly, we will take this comment into consideration when we plan future prospective trials to further evaluate the data from this present hypothesis-generating study

Point 5: The first paragraph of the discussion repeats a lot of the results around NLR and survival. This could be reserved to the results and thus would make for a more impactful discussion.

Response 5: The first paragraph of the discussion has been augmented to remove the repetition of the numerical results (lines 145-150). Additionally, the results have been reworded to include the statement that elevated NLR is associated with lower OS and PFS (lines 87-88). Results of the log-rank trend tests have been moved from their own separate paragraph to the paragraph corresponding to the value they were measuring (lines 81-99).

Point 6: Given the findings here, I think it could be impactful to make a statement of how to use NLR in stratifying patients, noting of course that larger data are necessary. If, for example, the trial that the authors suggest were to go forward, how would they suggest stratifying around NLR? At what numbers?

Response 6: We have further addressed stratification of NLR in the discussion, as seen on lines 240-242. Given the results of our study, which used a threshold NLR of 4.6 K/UL to separate groups, along with the results of Des, et. al. with their threshold of 5 K/UL and the threshold of 3.54 K/UL used by Dede, et. al.., we recommend future studies start with a threshold below that of Dede, et. al. to further ascertain at what point elevated NLR impacts outcomes.

Point 7: I disagree with the statement of evaluating NLR as it relates to immunotherapy in pancreas cancer in the discussion, because immunotherapy has no role for treatment of pancreas cancer save for rare cases of MSI or on clinical trial. 

Response 7: We appreciate this comment and agree that immunotherapy has no role in non-MSI high pancreatic cancers. Hence, we have deleted line 174 in the discussion section of the manuscript.